# From Offline to Real-Time Distributed Activity Recognition in Wireless Sensor Networks for Healthcare: A Review

**DOI:** 10.3390/s21082786

**Published:** 2021-04-15

**Authors:** Rani Baghezza, Kévin Bouchard, Abdenour Bouzouane, Charles Gouin-Vallerand

**Affiliations:** 1Département D’informatique et de Mathématique, Université du Québec à Chicoutimi, Chicoutimi, QC G7H 2B1, Canada; Kevin_Bouchard@uqac.ca (K.B.); abdenour_bouzouane@uqac.ca (A.B.); 2Departement of Information Systems and Quantitative Methods in Management, École de Gestion, Université de Sherbrooke, Sherbrooke, QC J1K 2R1, Canada; Charles.Gouin-Vallerand@USherbrooke.ca

**Keywords:** activity recognition, machine learning, offline, real-time, distributed, centralized, wireless sensor networks, streaming, concept drift, healthcare

## Abstract

This review presents the state of the art and a global overview of research challenges of real-time distributed activity recognition in the field of healthcare. Offline activity recognition is discussed as a starting point to establish the useful concepts of the field, such as sensor types, activity labeling and feature extraction, outlier detection, and machine learning. New challenges and obstacles brought on by real-time centralized activity recognition such as communication, real-time activity labeling, cloud and local approaches, and real-time machine learning in a streaming context are then discussed. Finally, real-time distributed activity recognition is covered through existing implementations in the scientific literature, and six main angles of optimization are defined: Processing, memory, communication, energy, time, and accuracy. This survey is addressed to any reader interested in the development of distributed artificial intelligence as well activity recognition, regardless of their level of expertise.

## 1. Introduction

Population aging comes with a set of problems that will have to be solved in the next decades. It is projected that nearly 2.1 billion people will be over the age of 60 by 2050 [1]. As disability [2], as well as dementia [3], rates increase with age, this segment of the population requires assistance in their daily lives.

By capitalizing on the use of technology for ambient assisted living, it is possible to extend the autonomy and the quality of life of patients suffering from mild degrees of dementia. Using technology may also allow us to reduce the ever increasing costs of healthcare [3] in terms of human and monetary resources. Smart environments and smart homes were not initially thought of as healthcare oriented, and the latter were described as “[…] a residence equipped with computing and information technology which anticipates and responds to the needs of the occupants, working to promote their comfort, convenience, security and entertainment through the management of technology within the home and connections to the world beyond” by Harper [4]. The initial vision was to increase convenience and comfort for any resident, but soon enough, researchers understood the potential of an environment filled with sensors to remotely monitor patients, and made use of these wireless sensor networks to assist them. Virone [5] presented an architecture that collects sensor data inside a smart home and stores it in a database so that patients can be remotely monitored, combining sensors, back-end nodes, and databases to track a patient inside their home.

In this paper, we explore the evolution of activity recognition from basic offline implementations to fully distributed real-time systems. The motivation for this review comes from the transition we are reaching in our research work in the use of ambient intelligence for assisted living in smart homes [6,7]. With the technological advancement of the Internet of Things, we are now beginning to investigate the distribution and real-time implementation of smart technologies, not only to improve current technologies used in smart homes, but to extend assistance to vulnerable users beyond smart homes, and into smart cities [8]. In order to reach that goal, it is necessary to have a good understanding and a wide overview of the field of activity recognition in smart homes and its evolution, from its infancy using binary sensors and simple learning rules, to more advanced, real-time distributed implementations in Wireless Sensor Networks using real-time distributed machine learning.

The switch from offline to real-time distributed activity recognition would allow us to bring timely assistance to semi-autonomous people by quickly detecting anomalies and being able to bring human assistance to patients when it is needed. Distributing the algorithm for activity recognition allows us to take advantage of the ever increasing capabilities of embedded devices and the IoT. With distribution, there is no need to run a costly distant server to process all the data, there is less need for long distance data communication, less privacy issues, and no single point of failure. However, there are still many challenges to overcome for real-time distributed activity recognition, which this survey attempts to cover.

This review is therefore aimed at researchers currently working in the field of offline activity recognition, looking for an overview of the possibilities for distributed and real-time approaches, as well as readers interested in the possibilities and challenges of the distribution of machine learning algorithms in wireless sensor networks. The main research questions this paper addresses, where each question is linked to a specific section are:What are the most commonly used methods, sensors, and algorithms to perform offline activity recognition, and what results have been achieved?What are the additional considerations and challenges when switching from offline to real-time activity recognition, and which architectures, algorithms, and communication protocols have been used?What are the current challenges of distributing real-time learning algorithms in wireless sensor networks, and which optimizations have been used to perform activity recognition in a distributed and real-time manner?

In the next section, we present the process used to collect the papers used for this review, starting with the database and the queries, to the inclusion and exclusion criteria for each subsequent section. We then present a quick survey of offline activity recognition systems, the main types of sensors used, the collected data, extracted features, algorithms used, and achieved performance. In the following section, we delve deeper into centralized real-time approaches for activity recognition, and we explore the network architectures and protocols used, the difference between local and cloud based implementations, and the new challenges rising with real-time machine learning for activity recognition. The fourth section focuses on the current state of the art in distributed real-time activity recognition, and highlights the challenges and drawbacks of a distributed approach. Finally, we discuss the next steps and research to be carried out in order to achieve high performance distributed real-time activity recognition, with a focus on embedded systems and the Internet of Things.

## 2. Methods

Due to the wide scope of this review, it is not possible to dive into the details of each component of activity recognition systems in offline, real-time, and real-time distributed systems using a quantitative approach in the way a classical systematic survey would. We have decided to opt for a narrative survey format [9] aimed at providing a qualitative understanding based on generalizations emerging from a large population, or a large body of research in this case.

For paper collection, we have used three different queries on Scopus, one for each section of this paper:“Activity Recognition” AND “Wireless Sensor Network” (211 results)“Activity Recognition” AND “Wireless Sensor Network” AND “Real-Time” (42 results)“Machine Learning” AND “Wireless Sensor Network” AND “Real-Time” AND “Distributed” (37 results)

The term “Activity Recognition” turned out to be too restrictive for the last section of this paper, which highlights the novelty of performing activity recognition in a real-time and distributed way at the current time. Scopus turned out to be the best database choice for a narrative survey as it includes publications from many different publishers, therefore allowing us to have a broad overview of the field of interest. Additional papers have been added when more resources were necessary to explore communication protocols or the distribution of specific machine learning algorithms.

For the offline section, the inclusion and exclusion criteria are the following:**Inclusion criteria:**-Privacy preserving sensors are used to collected data about one or several users for activity recognition purposes (cameras are excluded).-The paper presents explicit results for activity recognition, including relative results where different sensor setups are compared.**Exclusion criteria:**-Papers using sensorless or device-free approaches, such as Wi-Fi.-Papers focusing only on hypothetical architectures or sensor deployment methods (meta papers).-Papers focusing on energy consumption only.-Theoretical papers not providing any activity recognition results.-Papers using public datasets that were not collected by the authors during an experiment.-Any paper focusing on data reduction or compression methods only.

For the real-time section, the inclusion and exclusion criteria are the following:**Inclusion criteria:**-The activity recognition process has to be performed while the data is being collected, either directly after collection, or periodically throughout collection.-Papers using pre-collected data but simulating a real-time implementation are included as well.**Exclusion criteria:**-Any paper performing activity recognition with the use of the full, static dataset.

For the real-time distributed section, the inclusion and exclusion criteria are the following:**Inclusion criteria:**-Papers distributing machine learning algorithms across a wireless sensor network.**Exclusion criteria:**-Any experiment where no processing, data transformation, or classification task is performed on the sensor nodes.

As less research has been carried out in the distributed and real-time fields, the inclusion and exclusion criteria have been softened in order to include more papers. Each section of this paper serves as a building block for the following section, and to avoid any repetition, topics such as different sensor types are only addressed in the first section.

Many systematic surveys have been carried out in the field of activity recognition using sensors. Chen et al. [10] have reviewed sensor-based activity recognition, specifically comparing knowledge and data-driven approaches, and opening on the opportunities for complex and interleaved activity recognition. Lara et al. [11] have focused on activity recognition using wearable devices and proposed a two level taxonomy based on the use of supervised or semi-supervised learning algorithms, and an online or offline architecture. Wu et al. [12] have focused on sensor fusion for activity recognition, and Gravina et al. [13] have focused on data fusion specifically for body-sensor networks, where fusion strategies for heterogeneous data are investigated. Su et al. [14] have reviewed the use of smartphone sensors for activity recognition. Shoaib et al. [15] have addressed the issue of online activity recognition using smartphones, comparing the accuracy, energy, and resource consumption of different approaches used in the literature. Other surveys have focused more specifically on the learning algorithms, such as recent machine learning trends [16], transfer learning [17], or deep learning [18].

Since all of these papers provide a detailed analysis of the existing methods for activity recognition in a limited scope, we have decided to provide a narrative survey that would present a meta-analysis of the field of activity recognition as a whole, and its evolution towards real-time distributed approaches in the context of pervasive computing and the Internet of Things. The main limit of this study comes from its wide scope: Interesting papers in the field may have not been mentioned or discussed in detail. However, this is considered acceptable, as the aim of this survey is to highlight the high level challenges standing in the way of fully distributed, real-time activity recognition.

## 3. Offline Activity Recognition

The most straightforward way to perform activity recognition is to do it offline. In most applications, sensors are deployed in the environment or directly on the subject, data is collected, labeled, and fed to machine learning algorithms to perform activity recognition. In most cases, the sensors are organized in a Wireless Sensor Network (WSN), and the collected data is sent to a central computer. It is then labeled and used for training and testing of machine learning models. Offline activity recognition is useful to learn about behavioral patterns, such as learning Activities of Daily Living (ADL), or learning the way different individuals perform the same activity. In these cases, it is not necessary to have an instantaneous feedback from the system. For offline activity recognition, it is possible to use the whole dataset to train a classifier, as long as the hardware used is powerful enough. Otherwise, it is possible to split the data into smaller batches or parallelize training. Once the data is collected and labeled, it is easy to compare several machine learning models using various performance indicators. Offline activity recognition serves as a sandbox to find the most efficient algorithms for a specific use-case in a minimal constraint environment.

### 3.1. Sensors

A wide variety of sensors have been used for activity recognition applications. They are split into two main categories: Environmental and wearable sensors.

#### 3.1.1. Environmental Sensors

Environmental sensors are defined as static devices that capture physical measurements of properties of the environment they are deployed in. Many different kinds of sensors have been used in smart homes for activity recognition, from simple contact sensors to high resolution cameras.

Extensive research has been conducted for activity recognition in smart environments using binary sensors. Careful positioning of the sensors at key locations in the house allows for accurate activity recognition, as most activities are heavily location dependent: Teeth brushing happens in the bathroom and requires the tap to be turned on and off, whereas cooking happens in the kitchen, where the oven might be on. Passive infrared sensors (PIR) have been used to detect the presence or absence of motion by Ordonez et al. [19] and Samarah et al. [20] for ADL recognition. A combination of motion sensors and pressure plates have been used in [21] for visit recognition in elders’ homes. Despite the very simple sensors used, their combination allowed for a 85% visit detection rate. The same sensors complemented by reed switches, mercury contacts, and float sensors were used by Mittal et al. [22] for ADL recognition. Luo et al. [23] have used a smart combination of 9 PIR sensors with masking tape on some of the sensors, and were able to pinpoint the approximate location of a subject in a room based on the activation states of the sensors.

Binary sensors can only be in two states: 0 or 1. In many applications, a much higher resolution is needed to perform activity recognition. Digital sensors capture much more complex data, such as image, sound, temperature, distance, or humidity. A combination of force sensitive sensors, photocells, distance, sonar, temperature, and humidity sensors has been used in [24] to perform ambient assisted living in a multi-resident home. Indeed, the complex task of activity recognition for more than a single subject in the same house requires more fine-grained data, and binary sensors are no longer descriptive enough. Ching-Hulu et al. [25] have proposed the use of ambient-intelligence compliant objects (AICOs), which follows Weiser’s [26] vision of weaving profound technologies into the fabric of everyday life. AICOs act as a virtual layer added on top of physical objects, collecting data with the use of sensors without interfering with natural manipulation of the object. Du et al. [27] have improved the use of RFID tags on objects with a 3-stage activity recognition framework allowing us to capitalize on the usage state of different objects to infer the activity currently being performed and to predict the next activity in line. Galvan-Tejada et al. [28] have used sound data and a Random Forest for the development of an indoor location system relying on a human activity recognition approach: The activity performed by the user allows us to infer their location in the environment. Chapron et al. [29] have focused on bathroom activity recognition using Infrared Proximity Sensors (IRPS) in a cost-efficient system and have achieved an accuracy of 92.8% and 97.3% for toilet use and showering activities. Passive RFID tags have been used by Fortin-Simard et al. [7] to perform activity recognition in smart homes. RFID tags do not require any power source, and can be embedded everywhere in the environment. The authors have managed to detect anomalies in activity execution by the user related to cognitive impairment by relying solely on these tags. Outside of conventional sensors, low cost radars have been used by Cagliyan et al. [30], yielding good results for running, walking, and crawling recognition when the radar is facing the target.

Environmental sensors are interesting to researchers because they are not as intrusive or restrictive as wearable sensors. However, the collected data does not allow us to model the subject’s movements as accurately, and the activity recognition capabilities of environmental systems heavily relies on how distinct activities are from each other, and how location and time-dependent they are. Environmental sensors can still be used efficiently in real-time systems, especially for activity of daily living recognition, as these are often times highly coupled with the time and location of execution.

#### 3.1.2. Wearable Sensors

Wearable sensors are sensors integrated into wearable objects. The main types of wearable sensors used are accelerometers, gyroscopes, magnetometer, and RFID readers. Smartphones fall outside of that category, but since they can be used for their internal accelerometer and gyroscope while sitting in the user’s pocket, we discuss them as well in this section. Simpler wearable systems rely on a single accelerometer sensor, and others combine different sets of sensors on different parts of the body to fully capture the subject’s movements.

Sarcevic et al. [31] have used a combination of accelerometers, gyroscopes, and magnetometers on both wrists of the subject and achieved a 91.74% activity recognition accuracy on a set of 11 activities. Other experiments have used accelerometers on different parts of the body, such as the right part of the chest and the left thigh [32], on both wrists and the torso [33] with added RFID wristband readers in [34] to monitor human-object interaction. Raad et al. [35] have opted for the use of RFID tags on ankle bracelets for localization and tracking of elders with Alzheimer’s. In some cases, accelerometers and gyroscopes have been used to focus on a specific part of the body, such as the arm in [36], where 3 sets of accelerometers and gyroscopes are located on the wrist, elbow joint, and upper arm to recognize bowling action for the game of cricket. Ioana-Iuliana et al. [37] have used 2 accelerometers on the right hip and right lower leg to recognize posture-oriented activites, such as standing, sitting, walking, crawling with an accuracy of up to 99.2%. The location of the sensors on the subject’s body is a crucial part of the experimental setup, and determines what kinds of activities will be recognized. Sazonov et al. [38] have used a combination of accelerometers and pressure sensors on the sole of a shoe for posture and activity recognition with great success. Cheng et al. [39] have achieved an accuracy of more than 92% over a set of 5 activities using 4 accelerometers located on both ankles, the left thigh, and abdomen of the subjects.

More sensors have been combined with wearables, such as audio, temperature, humidity, and light sensors in [40] to recognize multi-user activities. Ince et al. [41] have combined a wrist worn accelerometer and environmental sensors to recognize brushing teeth, washing face, and shaving activities. Other works have used a smartphone’s built-in inertial sensors to perform activity recognition with good results, such as in [42,43]. They have also been combined with a wrist worn accelerometer in [44] to recognize user specific activities such as walking, jogging, running, cycling, and weight training. In this configuration, the smartphone not only collects data thanks to its inertial sensors, but it also acts as a storage node that receives data from the wrist worn accelerometer. Physical activity recognition has been performed using a hip worn accelerometer as well as a portable electrocardiogram (ECG) in [45]. Similarly, Liu et al. [46] have used a wearable ECG monitoring device coupled with a chest-mounted accelerometer for activity recognition with a 96.92% accuracy for coughing, walking, standing, sitting, squatting, and lying activities.

Wearable sensors allow for more precise movement recognition and are particularly suited for physical therapy applications. The main downside is that wearables can be intrusive, and in a real-life scenario, most people are reluctant to wear sensors on their body. In a healthcare, and assisted living context, elders and people suffering with dementia might simply forget to wear them. However, they allow us to detect dangerous situations such as falls more accurately than environmental sensors which is very useful in a real-time monitoring context. Additional challenges, such as communication between two nodes on opposite sides of the body for the needs of distribution, are covered later in this paper.

The combination of environmental and wearable sensors allows to achieve great results for activity recognition, and wearing several accelerometers on the body leads to better movement modeling. However, more data also means more computation, especially when accelerometers are set to collect hundreds of acceleration values per second. In this ocean of raw data, it is important to extract useful features to train machine learning models, to try and achieve the highest possible accuracy while still maintaining a good generalization to fit different users. The diagram below shows an overview of the main types of sensors used for activity recognition in the literature (Figure 1).

### 3.2. Labeling and Features

Regardless of the sensors used, once the raw data has been collected, it has to be labeled to be matched with the right activities in order to train supervised machine learning models. The next step is to extract features from the raw data in order to train and test the models.

#### 3.2.1. Labeling

For offline activity recognition, raw data is collected during the experiments and needs to be labeled to establish the ground truth. Labeling can either be done during the experiment by an observer taking notes on the activities performed at a specific time [47], or after the fact, using video footage of the experiment [43]. Another method that has been used is to provide the subject of the experiment with a form [21] or an activity labeling interface [24], which is known as active learning.

The first method allows researchers to have full control of the ground truth for their experiments, but it is very time consuming, especially on big datasets with several users spanning over several weeks, or months [19]. Labeling mistakes are also most likely to happen in this kind of dataset. Involving the subject in the labeling process allows researcher to offload some of that task to the user, even though the ground truth still has to be checked for errors and missing labels. In the context of healthcare and smart homes, the subjects might not always remember to fill in the activity forms all the time [21], because of dementia or amnesia, leading to missing labels. For shorter activities, it is unrealistic to expect the user to stop after every move to label the activity they have just performed.

Creating and labeling datasets to make them public and benefit the activity recognition community is therefore a tedious process, even if initiatives such as CASAS’s “Smart home in a box” [48] have helped in that regard. Mittal et al. [22] have explored the idea of using Formal Concept Analysis (FCA) and relying on the temporal context of activities to extrapolate activity labels from one source house to target houses. Activities from target houses can therefore be inferred with a certain confidence based on data from the source house. However, the system is not perfect, and there are many unclassified instances in new houses. If the layout and the habits of the residents change too much from one house to the other, this method does not perform as well.

The issue of finding ways to accurately and efficiently label activities in an automatic or semi-automatic way still remains, and it is even more important for the scientific community to address it for real-time activity recognition and activity discovery to allow for the development of more flexible systems.

#### 3.2.2. Features

Most offline approaches rely on supervised machine learning for activity recognition. A feature vector is associated with an activity label, which will be used for training as well as testing and determining the performance of the model. Data are continuously collected from the environment, whether it is through environmental or wearable sensors, and it is processed afterwards in the case of offline activity recognition. Before even deciding the kind of features to extract, the length of the window on which features will be computed has to be determined.

In the case of environmental sensor based activity recognition, 60 s time slices are used in most cases [19,22,24,49]. Tsai et al. [50] have worked on CASAS datasets and used a 5 min time slice instead. In the case of visitor recognition in elderly resident’s homes, the time slices were built around the average duration of a visit [21]. Samarah et al. [20] have used window sizes based on the number of sensor events from the beginning to the end of an activity, which is a commonly used technique for binary sensor based activity recognition, as binary sensors only collect data when their state changes. Using a combination of 9 PIR sensors, Luo et al. [23] have collected data using a frequency of 15 Hz, and computing features on 2 s windows with 50% overlap between consecutive windows.

When wearable sensors are used, data is collected continuously at a pre-defined frequency depending on the sensors capabilities, the resolution needed, and the storage capacity or transfer speed of the device. Commonly used accelerometer frequencies range from 10 Hz [37] for simple activities such as standing, sitting, or lying down to 50 Hz for most smartphone based systems [42,43], to 90 Hz [33], 125 Hz [31], 128 Hz [40], or 150 Hz [36] for higher resolution systems.

Once the raw data is collected, it is split into windows of either fixed or variable size. It is common to use a fixed window duration of 1 s, while the overlap between consecutive windows can vary a lot, from 15% [39] to the most commonly used 50% [37], to up to 87% overlap [33]. The higher the overlap, the heavier the computation is, but it allows us to not miss out on patterns that could increase activity recognition performance. Liu et al. [51] have provided a detailed analysis of different segmentation approaches and window sizes for fall detection using wearable sensors. They have identified the two main segmentation methods used in the literature: Sliding and impact-defined windows. While the first one simply splits the data in fixed size windows, the second approach centers the window around the fall. The authors have shown that sliding windows are more sensitive than impact-defined windows to size changes, and that the smaller the window is, the more energy efficient the system is. Once the windows are set, the feature vectors can be computed. In the case of environmental sensors, sequences of events and time domain features for digital sensors are used. For wearable sensors such as accelerometers or gyroscopes time domain and frequency domain features can be used.

Time domain features (TDF) are straightforward to extract by performing basic computation on the raw data contained in a specific window. The most commonly used TDF used are the mean absolute value, maximum, minimum, number of slope changes, number of zero crosssing, root mean square, standard deviation, Willison amplitude, and waveform length [31]. Frequency domain features (FDF) are computed after transforming the raw data from time domain to frequency domain, which entails heavier computations. One of the most popular technique is to used Fast Fourier Transform (FFT) to perform that transformation. Some of the most used FDF are the spectral energy, mean frequency, mean power, peak magnitude, peak frequency, and variance of the central frequency, all of which are detailed in [31] as well. Sarcevic et al. have also found in that paper that TDF tend to give slightly better results for activity recognition (91.74% accuracy against 88.51% for FDF).

The choice of the features to be extracted depends on the sensor type as well as the processing power of the system. Complex features do not always lead to better performance as mentioned above [31].

Additionally, when moving towards real-time or distributed activity recognition, processing time becomes a very valuable resource, and features have to be carefully selected in order to keep the most relevant ones, at the expense of a slight loss in accuracy. Zhan et al. [52] have managed to lower the computation burden by 61.6% while only losing 2.9% accuracy (92.5% to 89.6%) for activity recognition using a digital recorder by removing the overlap between consecutive windows. Indeed, they have extracted Mel Frequency Cepstral Coefficients (MFCC) from sound data. MFCC is defined as a representation of the short-term power of sound, and it is a frequency domain feature, which is computationally demanding to extract, leading to the window overlap reduction being necessary to reduce the computational impact of its extraction. There are two main ways to select features: Dimensionality reduction methods and feature transformation methods. The first method reduces the dimensionality of the vector by simply dropping the worst performing features, while the second method tries to map the initial features into a lower dimensional subspace [47]. Henni et al. [53] have presented an unsupervised, graph-based feature selection method, which falls into the latter category. Wang et al. [47] have used feature selection and observed that they could drop up to 14 features out of the original 19 and still achieve 88% accuracy for water drinking activity recognition. The most commonly used dimension merging methods are principal component analysis (PCA) and independent component analysis (ICA). Cagliyan et al. [30] have used PCA as a mean to estimate the contribution of each feature in order to find the optimal positioning for the radar sensor they have used. Lu et al. [25] have focused on explicit features (TDF), which are less computationally demanding than implicit features. After extraction, all features have been evaluated using two criteria: Time invariance and detection sensitivity, and only the best performing features have been kept, based on an empirically determined cutoff point.

Feature extraction and selection is a critical step of activity recognition using machine learning. Extracting too many features leads to a high computation cost, and the usefulness of each feature being greatly diminished (curse of dimensionality). Some features require more computation to extract, such as frequency domain features, and might not be suited for time-sensitive or limited resource systems. The diagram above shows a summary of the main FDF and TDF used for activity recognition (Figure 2).

### 3.3. Filtering and Outlier Detection

Since activity recognition is performed in a real-life environment, and is now extending to individual homes rather than labs, more noise and outliers have to be dealt with in the collected data. An outlier is generally defined as an observation that lies outside the overall pattern of a definition. In the case of activity recognition relying on sensor, we usually consider data points collected by faulty sensors to be outliers.

Noise is another issue that has to be dealt with. Sensors will collect more data than necessary for activity recognition, and a microphone might pick up constant background noise, or motion sensors can fire up when a cat is moving around in the house. This noise needs to be filtered in order to only feed relevant data to the machine learning algorithm.

Cagliyan et al. [30] have used a high pass filter to remove noise in the data collected by radar sensors. High-pass filters are commonly used to remove the unwanted DC component in accelerometer data, which corresponds to the contribution of gravity [33,36,37,41]. Low-pass filters have been used on accelerometer data in [40,43]. Nguyen et al. [54] have used a threshold based approach to eliminate noise from sound data. Using a threshold is a simple and effective method, however, the value of that threshold has to be determined empirically, and any deviation from the norm that was previously unseen in the data could break the threshold.

Most outlier detection techniques rely on clustering, based on the fact that by nature, outliers have a high dissimilarity compared to acceptable data. Outlier detection methods are split in 3 categories: Distance-based, density-based, and hybrid methods. Distance-based methods such as k-NN-DB compute the distance between data points and their neighbours to detect outliers, whereas density-based methods such as Local Outlier Factor (LOF) compare the local density of a data point to the local density of its nearest neighbors. Nivetha et al. [55] have described these methods and proposed a hybrid method that resulted in much better outlier detection performance. Ye et al. [56] also highlighted the fact that even though their system allows the detection and removal of outliers, it does not allow us to “add” missing values, which is another issue brought on by faulty sensors. Nivetha et al. [55] have shown an increase of up to 10% accuracy using their hybrid outlier detection method compared to using no outlier detection at all, and a 1 to 2% improvement compared to either distance or density based methods on their own. Noise filtering and outlier detection and removal methods are therefore a necessary step to increase activity recognition performance, even more so in a noisy environment, outside of a lab. The main steps of the process are illustrated in the diagram above (Figure 3).

### 3.4. Algorithms and Performance

Researchers have focused on machine learning for activity recognition in order to find the best performing algorithm in different contexts. Two comparison tables can be found for environmental sensor based approaches (Table 1) and wearable sensor based approaches, including smartphones (Table 2). When different datasets are used, the average of the accuracies on each dataset is computed and given in the table. The meaning for the abbreviations can be found in Abbreviations Section.

In smart homes using environmental sensors, Hidden Markov Models have been used alone [21,24,49] or combined with other models such as SVM and MLP to build a hybrid generative and discriminative model [19]. A Conditional Random Field (CRF) was used in [49], yielding better results than a Hidden Markov Model (HMM), with 95.1% accuracy instead of 91.2%. This difference in accuracy has been justified by the fact that a Conditional Random Field (CRF) is trained by maximizing the likelihood over the entire dataset, whereas HMM splits the data according the class labels, and optimizes the parameters for each subset separately. This leads to CRF adapting the possible class imbalance of the dataset and gives it a slight edge over HMM in terms of pure accuracy.

Decision Trees (DT) have not been the best performing algorithms for activity recognition in [24,47], but Tsai et al. [50] have achieved an 80% accuracy in predicting the next performed activity using DTs and a CASAS dataset. Luo et al. [23] have used a two-layer Random Forest (RF) with their combination of masked PIR sensors. The first layer of that RF focuses on the location and the speed of the user and achieves an 82.5% accuracy for activity recognition. Adding in the duration spent in a specific area of the room as an input to the second layer of the RF, the accuracy has been improved by 10%. As far as unsupervised machine learning goes, Bouchard et al. [57] have used an improved version of the Flocking algorithm on different datasets for activity recognition, and they have achieved an accuracy of up to 92.5% with 13,000 iterations of their clustering algorithm.

When it comes to wearable sensors, SVM have been used in [33,39,44,45,55,59] with accuracies ranging from 84.8% to 96%. Chiang et al. [32] have experimented with Fuzzy rules for postures and movement recognition with, respectively, 93% and 99% accuracy. Chawla et al. [44] have compared several algorithms (Artificial Neural Network (ANN), k-Nearest Neighbor (k-NN), Classification and Regression Tree (CART), and Support Vector Machine (SVM)) in terms of accuracy as well as training time for activity recognition using wearable sensors. They have found that although the ANN had the best accuracy 96.8%, it was the longest algorithm to train using all features with 13.64 s training time. k-NN, CART, and SVM have, respectively, achieved 96.2%, 95.3%, and 94.4% accuracy. Ordonez et al. [58] have used a deep neural network using a combination of convolutional and recurrent layers for gesture and mode of locomotion recognition, and they have achieved, respectively, 86.6% and 93% accuracy. One of the main advantages of this architecture is that there is no need for pre-processing, and the data from the sensor can be fed directly to the neural network.

### 3.5. Discussion

In this section, we have covered some of the main types of sensors, features, and main methods for outlier detection and machine learning models used. Sensors are mainly split between environmental and wearable sensors depending on the environment and the type of activities to be recognized: Environmental sensors are used in smart homes to recognize Activities of Daily Living (ADL), where sensor location and state are used to infer activities, whereas wearable sensors are mostly used to detect different postures and movements. For environmental sensors, sequences of activation are used, whereas Time Domain Features and Frequency Domain Features are extracted from accelerometer data. Threshold based noise filters and hybrid outlier detection are used to filter data before using it to train different models.

Offline activity recognition is useful to optimize models in order to achieve high accuracies, but they are not meant to be used in real-time application. In order to monitor elderly patients, real-time activity recognition is necessary, as immediate assistance might be required if any abnormal behavior is spotted.

## 4. Real-Time Centralized Activity Recognition

Real-time centralized activity recognition focuses on collecting data and recognizing activities in real-time, usually by aggregating data from the nodes of a wireless sensor network into a local computer or a distant server, and using machine learning algorithms to perform real-time or periodical classification of the performed activities. Switching from an offline to a real-time context is necessary when monitoring higher risk patients, or whenever the use-case requires an instantaneous feedback from the system. The sensors used are similar to the ones covered in the previous section: Environmental sensors, wearable sensors, smartphones, or any combination of these methods. As far as the architecture of the system goes, different network topologies can be used (mesh, star, partial mesh), as well as several main communication protocols (Wi-Fi, Bluetooth, ZigBee, ANT, LoRaWAN). Real-time systems sometimes provide a visualization tool to give feedback to the user or to the physician in the case of a healthcare application.

This section reviews real-time centralized activity recognition systems for healthcare, with an emphasis on the architecture of the systems as well as the issues and challenges of real-time and near real-time machine learning in a streaming context. We compare local and cloud-based approaches, and conclude on the state of the art in the field and the limitations of these systems.

### 4.1. Architecture and Communication

Efficient real-time activity recognition systems need to rely on a robust architecture. Data collected from each sensor node in the system needs to reach the central node in a timely manner. In this section, we review and compare the main communication protocols used for real-time activity recognition, as well as the most used network topologies. We also discuss data collection and storage issues in a real-time system.

#### 4.1.1. Communication Protocols

Wi-Fi, Bluetooth, ZigBee, and ANT (Adaptive Network Topology) are the main wireless technologies used for communication in Wireless Sensor Networks. We also cover the recently introduced LoRaWAN, which is aimed at long range, low energy communication. Cellular networks (GSM) can also be considered for smartphone based applications, but in the context of a WSN, they cannot be used to communicate with sensor nodes.

When it comes to newer technologies, even if 5G has not been used for activity recognition applications inside smart homes to the best of our knowledge, it seems very promising for IoT applications [60], and has been used for the promotion of unobtrusive activities and collision avoidance in the city [61]. Chettry et al. [62] have writen a comprehensive survey on the use of 5G for IoT applications. More recent Wi-Fi protocols, such as IEEE 802.11af have been used for healthcare applications to collect health data, such as body temperature and blood pressure from wearable sensors [63]. Aust et al. [64] have foreseen the upcoming challenge of highly congested classical wireless spectrums (2.4 GHz/5 GHz) due to the rapid advancement of the IoT, and they have reviewed the advantages of using sub 1 GHz Wi-Fi protocols, such as IEEE 802.11ah for industrial, scientific, and medical applications.

In terms of transfer speed, Wi-Fi offers the fastest solution. Newest standards such as 802.11ac advertise a maximum theoretical speed of to 1300 Mbps [65], with a maximum theoretical range of about 90 m outdoors, and 45 m indoors. Bluetooth 3.0, however, offers a maximum speed of 24 Mbps and a maximum theoretical range of 100 m. Starting from Bluetooth 4.0, a new standard was introduced as Bluetooth Low Energy (BLE), and was further improved in version 5.0, doubling its data rate [66]. The main goal of BLE is to reduce energy consumption in order to extend battery life of smartphones and wearables. However, the maximum transfer speed of BLE is 2 Mbps only, making it 12 times slower than regular Bluetooth. ZigBee has emerged in 2003 as a new standard particularly adapted to the Internet of Things (IoT) and sensor-based systems, with a focus on low latency and very low energy consumption [67]. LoRaWAN (Longe Range Wide Area Network) is a recently released (2016) network protocol built for LoRa compliant chips. It uses a star-of-stars topology and is advertised for long range, low energy consumption communication. Sanchez-Iborra et al. [68] have tested LoRaWAN’s maximal range and have found that packets could travel up to 7 km in an urban scenario, and 19 km in a rural scenario, thanks to the absence of obstacles in the way.

There are a lot of parameters to take into account for energy consumption, such as the type of packets used for Bluetooth, data transfer rate, sleep time, and transfer time for Wi-Fi. In each case, we picked the lowest consuming configuration presented in the papers in order to have an even ground for comparison. A comparative table can be found above (Table 3). In most cases, maximum range, transfer speed, number of nodes, and minimum power consumption cannot be achieved at the same time.

The choice of a protocol depends on the application and the main constraints of the system. In the case of a Wearable Body Sensor Network (WBSN), sensor nodes are powered through small batteries and need to rely on energy-efficient protocols such as BLE. However, if the number of nodes needed is high, such as in a smart home containing dozens of environmental sensors, ANT and ZigBee might be better suited. In applications where energy is not the main concern, but a high throughput is needed, such as in video based real-time activity recognition, Wi-Fi or even wired alternatives will most likely be the best choice. Because of its long range and low speed properties, LoRaWAN is more suited for periodic, low speed exchanges over long distances, but it is not particularly suited for WBSN applications.

#### 4.1.2. Topology

Real-time activity recognition systems collect data continuously from environmental or wearable sensors. Regardless of the type of sensors used, each sensor node has to send data back to the central node, which is usually a computer or a smartphone.

Suryadevara et al. [69] have used environmental sensors with ZigBee components organized in a star topology to determine the wellness of inhabitants based on their daily activities. The main advantage of a star topology is that each end node only has to know how to reach the central node, which simplifies communication. However, each sensor node has to be in reach of the central node, which might not always be the case for activity recognition in extended spaces. The central node also represents a single point of failure. Cheng et al. [39] have presented an architecture in which the failing central node is replaced with the next most powerful node available so that the system can keep running.

In order to improve the flexibility of their system, Suryadevara et al. [70] have experimented with a mesh topology using XBee modules to forecast the behavior of an elderly resident in a smart home. A partial mesh topology is used, where 3 relay nodes can forward data from the end nodes to the central coordinator. The reliability of the network has been shown to be above 98.1% in the worst case scenario, with 2 hops between the end node and the central coordinator.

Baykas et al. [71] have compared the efficiency of star and mesh topology for wireless sensor networks and shown that in wide area networks, mesh topology networks require 22% more relay nodes to support sensor traffic as reliably as the star topology equivalent. The nodes in a mesh topology also need 20% more bandwidth to deliver data in a timely manner.

ZigBee and ANT also support cluster tree topologies, allowing the use of hubs that receive data from their end nodes and forward it to the other nodes of the network. This topology can be useful when it is not possible to connect every end node to a single central node directly, or when the use of intermediary relay nodes allows us to handle traffic in a more efficient way. Altun et al. [72] have used a relay node on the chest to forward data from the left wrist end node to the central XBus master node located on the user’s waist.

Büsching et al. [73] have presented a disruption tolerant protocol to ensure that nodes in wireless body area networks could switch from online to offline storage should their connection to the central station be interrupted. After the connection is lost, each node will store the collected data on an SD card until the connection is restored. When the node is back online, it has to go through a synchronization procedure in order to send the backlog of data in the correct order to the central station.

In the context of activity recognition in a single room or a small house, it seems preferable to rely on a star topology as long as all the end nodes are in range of the central coordinator to ensure optimal reliability with the least amount of nodes. The figure below (Figure 4) shows the main types of topologies used in the literature.

#### 4.1.3. Data Collection

In real-time systems, data can either be collected from sensor nodes at a fixed time interval with the associated timestamp, or in the case of binary sensors, the new state of the sensor can be collected whenever it changes. Nguyen et al. have used a fixed time interval of 1 min in their real-time activity recognition system in an office [54]. Every single minute, each sensor sends its state to the base station, that then determines the activity that is being performed by the worker in the office based on a set of binary rules.

However, this method can lead to a lot of data being collected and stored, as well as redundant data when the sensors stay in the same state for an extended period of time. Suryadevara et al. [69] have instead opted to use an event-based approach for data collection, where the data collected from the sensors is only stored if the most recent state of the sensor differs from the last stored state. This allows us to store much less data, and to discard any repetitive data. This approach only works for binary, or to a certain extent, digital sensors. In most wearable sensor-based systems, data has to be continuously collected, as sensors such as gyroscopes or accelerometers usually operate in the 10–100 Hz sampling frequency range, and every single data point has to be stored to extract accurate features for activity recognition [74].

In order to improve the efficiency of the system, Chapron et al. [75] have opted for a smart data compression based on the range of definition of data collected by the accelerometer. They have noticed that full float precision was not required to store accelerometer, magnetometer, and gyroscope data, and they have successfully compressed data so that each packet could contain the complete information from the 9-axis IMU and fit in a single BLE characteristic.

After choosing the right sensors based on the activity recognition task, it is critical to build a suitable architecture and opt for an appropriate communication protocol for real-time application. These choices should allow for a fast, reliable, and energy efficient system. Data also has to be handled and stored in a way that minimizes storage and processing requirements of the central station.

### 4.2. Local and Cloud Processing

In a centralized activity recognition system, data is collected from sensor nodes and sent to a central station. That central station can either be a local computer, located in the lab or the house, or it could be a distant server that receives the data, processes it, and uses it for activity recognition. The main advantage of cloud computing is to offset the processing and complex calculations to powerful distant machines [76]. It also provides a very high availability, and a single set of machines can be used for different applications by splitting their processing power and resources amongst different tasks. More recently, as another sign of the trend shifting from centralized to distributed and pervasive computing, the Fog computing model has been developed, where some of the processing is distributed to intermediary nodes between the end device and the distant servers [77].

#### 4.2.1. Local Processing

Smartphones can also be used for local activity recognition, such as the system built by He et al. [78] relying on sensors wirelessly connected to a gateway, connected to a smartphone via USB. Kouris et al. [79] have used 2 body-worn accelerometers and a heart rate monitor sending data to a smartphone using Bluetooth. Similarly, Zhang et al. [80] have used accelerometers and gyroscopes communicating with a smartphone using Bluetooth. Biswas et al. [81] have used an accelerometer on the dominant wrist of a subject to perform arm movement recognition. The data is sent from the accelerometer to a local computer, which then transfers it to a FPGA board using a RS232 cable. The whole processing and machine learning is performed by the FPGA that is directly linked to the local computer.

#### 4.2.2. Cloud Processing

Cloud processing is particularly suited to healthcare applications, as it allows to build a system where the data and results of activity recognition are stored online. These results can then be accessed by the patients directly, as well as the physicians, and any abnormal behavior can be remotely spotted in real-time.

Ganapathy et al. [82] have presented a system made of a set of body sensors, such as blood pressure, heart rate, respiration rate, ECG, and SPO2 sensor sending data to a smartphone using Bluetooth. These data are then wirelessly sent to a distant server. Predic et al. [83] have used the inertial sensors of a smartphone, combined with air quality data collected through environmental sensors. All of the data is then sent to a cloud to be stored and processed. Mo et al. [84] have used a smartphone to collect data from body worn sensors with energy harvesting capabilities, and to send it over to a distant server for processing and activity recognition. Serdaroglu et al. [85] have used a wrist worn watch with a built-in accelerometer to collect data in order to monitor patients’ daily medication intake. The data is sent wirelessly from the watch to a gateway, connected to a computer via USB, before being sent over to a web server. The monitoring application is cloud-based, and both patients and doctors can access it. Khan et al. [86] have also used on-body worn accelerometers with an emphasis on the accurate positioning of the nodes on the subject’s body in order to improve the activity recognition accuracy. They have used energy-based features and a cloud-based architecture to perform activity classification. Fortino et al. [87] have presented the CASE (Cloud-Assisted Agent-Based Smart Home Environment) system where data is collected from both environmental and wearable sensors. Environmental sensors send data to a base station, such as a Raspberry Pi or a Beagle Bone, and wearable sensors send collected data to a smartphone. These data are then forwarded to the cloud-based architecture for processing.

Cheng et al. [39] have compared the efficiency of a local Python machine learning script running on a local computer and a distant Matlab algorithm running on the cloud. They have found that the accuracy of SVM seems to be better on the cloud (97.6% vs. 95.7%) and the accuracy of k-NN is better using the local Python script (97.9% vs. 95.9%). As cloud-based solution often times require the use of proprietary software and additional costs to rent storage space and processing power, Cheng et al. have decided to rely mainly on the localized algorithm, and only use the cloud algorithm if the local one fails.

The context of application of a real-time activity recognition system is the main criteria to determine whether to rely on a local or cloud based architecture. In the case of a healthcare system where remote monitoring from the doctors is necessary, a cloud architecture seems to be more suited. It also allows the patient themselves to access their history, as well as family members, either from their computer or an application on their smartphone or tablet. However, cloud-based systems are usually more costly, because of the monthly fee and the maintenance needed, as well as the possible use of proprietary software, and the difference in activity recognition accuracy alone usually cannot justify the extra cost.

The diagram below summarizes the main architectures used in the field of real-time activity recognition (Figure 5). The dashed line makes the difference between local and cloud based systems in order to compress the representation. The first approach uses sensor nodes wired to a gateway node that communicates with a computer. This approach is generally used with wearables in Wireless Body Sensor Networks (WBSN), where nodes are attached on different parts of the user’s body, and a gateway node is used to store and transfer the collected data to a local station that could be connected to the cloud. This approach allows for a faster and more reliable communication between the sensor nodes and the gateway, but it makes the system less practical and more invasive as the number of nodes increases because of all the wires involved. The second approach uses wireless communication between the sensor nodes and the gateway node. This approach is generally used with environmental sensors, as using wires between every single sensor node and the gateway node would be impractical in a smart home. The gateway node is directly plugged into the central station, which could process the data locally or send it over to a cloud server [85]. In the last presented architecture, all of the sensor nodes are sending data wirelessly to a smartphone, acting as a gateway node, sending data to the local station. In some applications, the smartphone sends data over directly to cloud servers [82]. Two different gateway nodes can also be used in the same system [87].

### 4.3. Real-Time Machine Learning

Supervised machine learning algorithms are the most commonly used class of algorithm in activity recognition. Data are collected, manually labeled, and used to train a model. Once the model has been trained, it can be used to perform activity recognition. In real-time applications, the simplest approach is to collect data and train the model offline. Once the model is trained, it can be used online to classify new instances and perform activity recognition. It is also possible to train some models in real-time by using smaller batches from the continuous flow of incoming data in a streaming context. In both cases, the main challenge is to label data and evaluate the true performance of an algorithm. If activities are performed in real-time without a human observer to provide the ground truth for verification, the accuracy of the model cannot be evaluated for supervised machine learning based systems.

In this section, we explore some proposed techniques of automatic activity labeling, highlight the main challenges of real-time activity in a streaming context, and review some of the approaches that have been applied to activity recognition applications.

#### 4.3.1. Activity Labeling

In offline machine learning systems, labeling is done by hand, either directly by an observer, after the experiment by using video recording, or by the subject themselves with the help of a form or an application.

Naya et al. [88] have provided nurses in a hospital with a voice recorder to allow them to describe the activity they were performing in real-time. However, the recording still has to be interpreted by a human in order to turn it into a usable activity label that can be used to train a machine learning model. In real-time systems, activity recognition can either happen in a closed or open universe. In a closed universe, a complete set of activities is defined from the start [54], and any new collected data forming a feature vector will be classified in one of the classes of this set, or remain an unclassified instance in some cases [89]. In an open universe, new activities can be discovered as the system runs, and irrelevant activities can be discarded.

Suryadevara et al. [69] have created a table mapping the type and location of each sensor, as well as the time of the day to a specific activity label. Using this technique, they have achieved 79.84% accuracy for real-time activity annotation compared to the actual ground truth collected from the subjects themselves. This approach is efficient in systems where a single sensor or a set of sensor can be discriminative enough to narrow down the activity being performed. In this closed context, no new activities are discovered. Fortino et al. [87] have used the frequent itemset mining algorithm Apriori to find patterns in collected data. Events represented by quadruples containing a date, a timestamp, the ID of a sensor and its status are recorded. These events are then processed to form a list of occupancy episodes in the form of another quadruple containing a room ID, a start time, a duration, and the list of used sensors. The idea behind this quadruple is to automatically represent activities that emerge as a function of the sensors firing, the time and duration of their activation, as well as the room in which they are located. Apriori is used to find the most frequent occupancy episodes, which are then clustered. Clusters can change throughout the system’s lifecycle, and each cluster acts as the representation of an unknown activity. The name of the activity itself cannot automatically be determined, and human intervention is still necessary to properly label it. This method is useful when there is a high correlation between time, location, and the observed activity.

Through active learning [90], it is possible to provide the user with an interface that allows them to give feedback over the automatic label suggested by the system. If the label is correct, the user specifies it is, and the new learned instance is added to the base of knowledge. Semi-supervised learning [91] can be used together with active learning to compare activity annotation predictions with the ground truth provided directly by the user. The model is first trained with a small set of labeled activities. Classification results for unknown instances are then checked using active learning, and added to the training set if they have been correctly classified, thus progressively allowing the training set to grow, and making the model more accurate and versatile in the case of activity discovery.

The smaller the set of activities to classify is, the easier it is to link a sensor to an activity. However, a sensor could be firing, letting us know that the sink is running, but it would be impossible to determine if the subject is washing their hands, brushing their teeth, shaving, or having a drink. The more complex the system gets, and the more sensors are added, the more difficult it becomes to establish a set of rules that link sensor activation to human activities. Most real-time systems have to be periodically re-trained with new ground truth in order to include new activities, and take into account the fact that the same activities could be performed in a slightly different way over time.

#### 4.3.2. Machine Learning

There are two main distinctions to be made when it comes to real-time machine learning: Real-time training and real-time classification. The latter represents the simplest case of real-time machine learning: A model is trained offline with a fixed dataset, in the same way offline activity recognition is performed, and it is then used in real-time to classify new instances. For most supervised learning based methods, classification time is negligible compared to training time. Models that require no training, such as k-NN require a higher classification time. Nugyen et al. [54] have used binary rules that map sensor states to an activity label to classify new instances in near real-time (5 min time slices). Other straightforward approaches use real-time threshold based classification [92] or a mapping between gyroscope orientation and activities [81].

Cheng et al. [39] have used both local and cloud based SVM and k-NN implementation for real-time activity recognition on an interactive stage where different lights are turned on depending on the activity of the speaker. k-NN [93] requires no training time, as it relies on finding the k nearest neighbours of the new data instance to be classified. However, in its original version, it has to compute the distance between the new instance and every single data point in the dataset, making it a very difficult algorithm to use for real-time classification.

Altun et al. [72] have compared several algorithms in terms of training and storage time for activity recognition using wearable sensors. Algorithms such as Bayesian Decision Making (BDM), Rule Based Algorithm (RBA), decision tree (DT), K-Nearest Neighbor (k-NN), Dynamic Time Warping (DTW), Support Vector Machine (SVM), and Artificial Neural Network (ANN) are trained using 3 different methods: Repeated Random Sub Sampling (RRSS), P-fold, and Leave one out (L1O) cross validation. Using P-fold cross validation, DT has been shown to have the best training time (9.92 ms), followed by BDM (28.62 ms), ANN (228.28 ms), RBA (3.87 s), and SVM (13.29 s). When it comes to classification time, ANN takes the lead (0.06 ms), followed by DT (0.24 ms), RBA (0.95 ms), BDM (5.70 ms), SVM (7.24 ms), DTW (121.01 ms, taking the average of both DTW implementations), and k-NN in last position (351.22 ms). These results show that DT could be suited for both real-time training and classification, as it ranks high in both categories. Very Fast Decision Tree (VFDT) based on the Hoeffding bound have been used for incremental online learning and classification [94]. Even though ANN requires the most training time, it performs the quickest classification out of all the algorithms compared in this paper, and could therefore be used in a real-time context with periodic offline re-training. The ability of neural networks to solve more complex classification problems and automatically extract implicit features could also make them attractive for real-time activity recognition. These results were obtained for classification of 19 different activities in a lab setting, after using PCA to reduce the number of features.

Song et al. [95] have explored online training using Online Sequential Extreme Learning Machine (OS-ELM) for activity recognition. Extreme Learning Machine is an optimization learning method for single-hidden layer feedforward neural network introduced by Huang et al. [96]. This online sequential variation continuously uses small batches of newly acquired data to update the weights of the neural network and perform real-time online training. ELM is particularly adapted to online learning as it has been crafted to deal with the issue of regular gradient-based algorithms being slow, and requiring a lot of time and iterations to converge to an accurate model. ELM has been shown to train NN thousands of times faster than conventional methods [96]. OS-ELM has been compared to BPNN [95], and has achieved an average activity recognition rate of 98.17% accuracy with a training time of about 2 s, whereas BPNN stands at 84.56% accuracy with a 55 s training time. This result show that neural networks could be a viable choice for online training as well as online classification, as long as the training procedure is optimized.

Palumbo et al. [97] have used Recurrent Neural Networks (RNN) implemented as Echo State Networks (ESN) coupled with a decision tree to perform activity recognition using environmental sensors coupled with a smartphone’s inertial sensor. The decision tree constitutes the first layer, and possesses 3 successive split nodes based on the relative value of collected data. Each leaf of this decision tree is either directly an activity, or an ESN that classifies the instance between several different classes. ESN is a particular implementation of the Reservoir Computing paradigm, that is well suited to process streams of real-time data, and requires much less computation than classical neural networks. The current state of a RNN is also affected by the past value of its input signal, which allows it to learn more complex behavior variations of the input data, and is especially efficient for activity recognition, as the same recurrent nature of certain behaviors can be found in human activity recognition (present activities can help inferring future activities). On a more macroscopic scale, Boukhechba et al. [98] have used GPS data from a user’s smartphone, and an online, window-based implementation of K-Means in order to recognize static and dynamic activities.

In a streaming context with data being collected and used for training and classification, several issues can arise. Once the architecture and communication aspects have been sorted as described in the previous sections, the nature of the data stream itself becomes the issue. As time goes on, it can be expected that data distribution will evolve over time and give rise to what is referred to as concept drift [99]. Any machine learning model trained on a specific distribution of input data would see its performance slowly deteriorate as the data distribution changes. As time goes on, new concepts could also start appearing in data (new activities), and some could disappear (activity no longer performed). These are called concept evolution and concept forgetting. The presence of outliers also has to be handled, and any new data point that does not fit in the known distribution does not necessarily represent a new class.

Krawczyk et al. [100] have reviewed ensemble learning based methods for concept drift detection in data streams. They have also identified different types of concept drift such as incremental, gradual, sudden, and recurring drift. Ensemble learning uses several different models to detect concept drift, and to re-train a model when concept drift is detected. The freshly trained model can be added to the ensemble or replace the currently worst performing model if the ensemble is full. Concept drift is usually detected when the algorithm’s performance starts to drop significantly and does not return to baseline. Some of the challenges of concept drift detection are to keep the number of false alarm to a minimum, as well as to detect concept drift as quickly as possible. Various methods relying on fixed, variable size and a combination of different window sizes have been described in [100]. The figure below illustrates the process of concept drift detection and model retraining (Figure 6).

Additionally, in high speed data-streams with high data volumes, each incoming example should only be read once, the amount of memory used should be limited, and the system be ready to predict at any time [101]. Online learning can either be performed using a chunk-by-chunk or one-by-one approach. Each new chunk or single instance is used to test the algorithm first, and then to train it, as soon as the real label for each instance is specified. This comes back to the crux of real-time activity recognition, which is the need to know the ground truth as soon as possible to ensure continuous re-training of the model. Ni et al. [102] have addressed the issue of dynamically detecting window starting positions with change point detection for real-time activity recognition in order to minimize necessary human intervention the segment data before labeling it.

In limited resources environments, such as when machine learning is performed on smartphone, a trade-off often has to be found between model accuracy and energy consumption as shown by Chetty [103] and He [78]. This is especially true for distributed real-time processing, which we cover in the next section.

### 4.4. Discussion

In this section, we have covered the main steps to follow in order to build a real-time activity recognition system, starting from the physical architecture, to the communication protocols, data storage, local and remote processing, activity labeling, machine learning in data streams, and the interaction with different actors of the system. Types and location of sensors have been covered in offline activity recognition section.

Many challenges still have to be faced for real-time activity recognition, such as real-time activity labeling in real-time, adaption to concept drift and evolution, timely online training and classification, as well as memory efficient algorithms. We have reviewed several real-time algorithms and highlighted the importance of semi-supervised and unsupervised approaches to reduce necessary human intervention.

With portable and embedded devices becoming more and more powerful by the year, the Internet of Things becoming the new standard, and the convenience of pervasive computing, it seems natural to transition from centralized to distributed activity recognition, and to explore the new research challenges and opportunities that rise with it.

## 5. Real-Time Distributed Activity Recognition

With IoT boards becoming smaller and more powerful, distributed activity recognition is the next logical step to weave technology more profoundly in everyday life. In an ideal case and a fully distributed system, no node is essential and there is no single point of failure, as opposed to centralized activity recognition that relies on a local computer or a distant server for processing and classification. Distributed activity recognition also allows the creation of fully autonomous systems that do not rely on an external internet connection to keep performing their tasks. A local system also implies a higher degree of privacy, as no single node stores all of the data.

However, distribution comes with a whole new set of problems, especially in a real-time context. In this section, we first cover the on-node processing aspects of distributed activity recognition, then we move on to communication between nodes, and we conclude on distributed machine learning. In each section, we review some of the methods that have been found in the literature and we identify the main considerations to take into account when building a distributed activity recognition system.

### 5.1. On-Node Processing

The main difference between centralized and distributed systems is the nodes’ ability to perform some processing before communicating with other nodes. In a distributed system, basic feature extraction can be performed on low-power nodes, which allows it to save a lot of energy on communication, as features extracted on a data window are generally more compact than raw data [104].

However, because of the nodes’ limited processing power, feature extraction has to be as efficient as possible. Lombriser et al. [105] have compared features in terms of the information gain they provide to the classifier versus their computational cost, and they have found that the mean, energy, and variance were the most efficient features for on-body activity recognition with DT and k-NN, using accelerometer data.

Roggen et al. [106] have used a limited memory implementation of a Warping Longest Common Subsequence algorithm to recognize basic movement patterns. In order to reduce necessary processing power and code space, the algorithm relies on integer operations rather than floating point operations. They have implemented the algorithm on a 8-bit AVR microcontroller and a 32-bit ARM Cortex M4 microcontroller, and they have shown that a single gesture could be recognized using only 0.135 mW of power, with the theoretical possibility of recognizing up to 67 gestures simultaneously on the AVR, and 140 on the M4. Because of the nature of the algorithm they have used, Roggen et al. have chosen to not extract any features and find the Longest Common Subsequence directly using raw data. This leads to more complex movement recognition being less efficient, but it reduces the computational burden on the nodes.

Lombriser et al. [105] have observed that bigger window sizes and wider overlaps yielded higher activity recognition accuracy, but came with a higher computation burden as well. They have settled for a middle ground with 2.5 s windows and a 70% overlap. Roggen et al. [106] have reached similar conclusions as they have found that the shorter the sequence to match is, the lighter the computational burden is at the expense of losing specificity on activities to be recognized. Indeed, shorter patterns are less discriminative, and the shorter they are, the further away they are from describing a full gesture.

Jiang et al. [107] have focused on rechargeable sensor networks, using RF waves from RF readers to supply energy to Wireless Identification and Sensing Platform tags (WISP tags). They have compared different scheduling methods to distribute energy amongst the nodes at runtime in order to achieve the highest Quality of Monitoring (QoM) possible, defined by the ratio between occurred and captured events. They have found that using a hybrid method with scheduling on both the reader and the WISP tags allowed for the best QoM results.

In order to save energy whilst still maintaining a high activity recognition accuracy, Aldeer et al. [108] have used a low energy, low reliability motion sensor (ball-tube sensor) together with a high energy, high reliability sensor (accelerometer) for on-body acitivity recognition. The idea is to put the accelerometer to sleep when the user is mostly static, and to use cues from the ball-tube accelerometer to detect the beginning of a movement in order to enable accelerometer data collection. This allows to reduce energy consumption by a factor of 4 during periods where the user is static.

### 5.2. Communication

Communication is one of the main sources of energy loss in WSN together with high sampling sensors. Lombriser et al. [105] have found that radio communication accounted for 37.2% of the total node’s energy consumption, which is the second biggest energy consumption after the microphone (45.8%) and far more than the CPU (3.5%). It is therefore crucial to find ways to save more energy by communicating efficiently and minimizing packet loss as much as possible.

Considerable packet loss can be experienced in WBSN when a node tries to communicate with another node that is not in Line of Sight (LOS), such as shown in [105] where the transmission rate from a node on the waist to a node on the ankle falls to 78.93% when the subject is sitting down. Zang et al. [109] have designed the M-TPC protocol to address this issue in the specific case of the walking activity with a wrist worn accelerometer and a smartphone in the opposite pocket. They have noticed a negative correlation between acceleration values picked up by accelerometers and packet loss. Indeed when the body of the subject stands between the node and the smartphone, more data loss is experienced. M-TPC is designed to send packets of data only when acceleration is at its lowest, meaning that the subject’s arm is either in front or behind their body, at the peak of the arm’s swinging motion. This protocol allows to reduce transmission power by 43.24% and reduces packet loss by 75%.

Xiao et al. [110] have used Sparse Representation based Classification with distributed Random Projection (SRC-RP) to recognize human activity. By randomly projecting data to a lower dimensional subspace directly on the nodes, they can efficiently compress data and save on transmission costs while still maintaining a high activity recognition accuracy. With a 50% data compression rate, they have achieved an activity recognition accuracy of 89.02% (down from 90.23% without any compression) whilst reducing energy consumption by 20%. The authors have compared RP to PCA, and found that RP yields a very close accuracy, whilst being less computationally expensive than PCA, as well as being data independent.

De Paola et al. [111] have used a more centralized approach, but they have explored the issue of optimal sensor subselection in order to save energy in a WSN. Each node collects data, and a central Dynamic Bayesian Network is used to perform activity recognition based on environmental sensors in a smart home. The information gain of each sensor is computed, and if the state of the system is not expected to change much, the least relevant sensors are set to sleep mode to save energy. Using this adaptive method, they have achieved 79.53% accuracy for activity recognition, which is similar to the accuracy using all sensors, but with a power consumption three times lower.

### 5.3. Machine Learning

In an ideal distributed activity recognition system, processing and classification tasks should be equally split between all the nodes. However, most approaches in the literature still rely on a central node to perform the final classification steps. There are different levels of distribution, starting from minimal on-node processing, to partial on-node classification, to fully distributed classification. A summary of the different degrees of distribution observed in the reviewed literature can be found below (Figure 7).

Farella et al. [112] have used on-body accelerometers for activity recognition. Their approach is mostly centralized, but they have used a lookup table that matches specific accelerometer values to the orientation of the node. This table is directly implemented on the nodes and allows a first basic pre-processing step to be carried out locally. Bellifemine et al. [113] have presented an agent-oriented implementation of the SPINE framework, designed to facilitate signal processing based tasks, such as activity recognition. Each node is able to perform basic feature extraction for accelerometer values based on a split function (mean, minimum, maximum for each axis), and uses an aggregation function to group up the features and send them to a central station.

Roggen et al. [106] have used majority voting in their LM-WLCSS system to find the best overall match among the nodes. Zappi et al. [114] have trained a HMM on each node and used a central node to weigh in the contribution of each node and perform classification using either majority voting or a Naive Bayesian fusion scheme. Bayesian voting proved more efficient, especially when faced with noise in the data, as the contribution of noisy nodes is reduced. Palumbo et al. [115] have used wearable and environmental sensors for distributed activity recognition using Received Signal Strength (RSS) values and Recurrent Neural Networks. In the distributed implementation for the walking activity, each node collects acceleration and RSS data, discards bad packets and sends data to the feature extraction module. Each node uses these features to make an activity prediction using the Echo State Network. Each prediction is sent to a gateway that performs majority voting as the final classification step. They have achieved a 91.11% accuracy for two activities (standing up and sitting down) using a centralized version of the algorithm. The authors have also shown that their algorithm only requires 2kB of RAM, which makes it suitable for embedded applications.

Wang et al. [116] have used a 2-layer classification system where gesture recognition is performed directly on the sensor nodes worn on the body and around the wrists, and sent to a central smartphone that performs higher level activity recognition. On each node, K-Medoid clustering is used to find templates for each gesture. At activity recognition time, Dynamic Time Warping (DTW) is used to find the closest matching sequence to the training set using a 1 s sliding window on incoming data. Each gesture has a different pattern, and the body and wrist nodes try to match different patterns. All sensors send the results of this initial classification to the central node that merges it into a bitmap. Emerging Pattern (EP) is then used to recognize higher level activities. The average accuracy of the system is 94.9%, however, it is not as suited for interleaved activity recognition because of the importance of pattern temporality. They have also shown that performing the first stage of classification directly on the nodes allowed for 60.2% savings in energy consumption when compared to sending raw data for central processing.

Atalah et al. [117] have used a 2-stage Bayesian classifier using data from an ear-worn sensor combined with environmental sensors. The ear-worn sensors first classifies the performed activity into a different category of activities based on the heart rate of the user. This first estimation is sent to the central node that receives additional data from the environmental sensor and uses it to perform the second classification stage. The authors have found that using the combination of both types of sensors allowed to reduce class confusion rates by up to 40% for some subjects, rather than relying only on wearable sensors.

Amft et al. [104] have also used a 2-stage classifier in their distributed user activity sequence recognition system. The first layer of classification happens directly on the nodes where atomic activities are recognized by finding the closest match to a known pattern in the data. These atomic activities are organized in an alphabet, and different sequences of atomic activities correspond to a composite activity. Using body-worn accelerometers and environmental sensors in a car assembly scenario, the authors have identified 47 atomic and 11 composite activities, and achieved a 77% activity recognition accuracy. They have also observed a 16% data loss when sending raw data from the nodes to a central coordinator, which highlights the advantage of on-node processing for reduced data transmission in distributed systems.

Fukushima et al. [118] have fully distributed a Convolutional Neural Network (CNN) in a WSN. Each node of the network is responsible for the computations of all the layers for a specific unit of the CNN, the same way CNN are used for image recognition, where a unit is a pixel or a group of pixels, except each node collects temperature or motion data in the two presented experiments. The CNN consists of an input layer, T hidden layers, a fully-connected layer and an output layer. Each hidden layer contains a convolutional layer and an optional pooling sublayer. Each convolutional layer has K filters. To ensure a lighter computational load, the number and size of the filters are limited. When a node receives all the necessary inputs from neighboring nodes, it goes through the layers, computes the output, and advertises the output for the units in the following layer. When a sensor node obtains ground truth, it begins the distributed backpropagation process, where each unit updates its weights based on the propagation from the previous units, therefore, there is no global optimization of the weights. MicroDeep has achieved 95.57% accuracy for temperature discomfort recognition, whereas a standard CNN running on a computer has achieved 97.1% accuracy. Using optimal parameters, MicroDeep achieves the same accuracy as the standard CNN, but its communication cost is multiplied by 8 as opposed to using a feasible configuration. Since the CNN forms a 2-D grid, there might be cells without an associated sensor. These cells are referred to as holes, and the authors have shown that they can still maintain a 94.4% accuracy even with 20% holes. This is an example of a fully distributed, real-time activity recognition system, which relies on a high number of sensor nodes to distribute processing.

Bhaduri et al. [119] have presented a distributed implementation of a Decision Tree in a P2P network using misclassification error as a gain function. Each node has a set of training examples and the aim of the algorithm is to select the best split based on each node’s decision, using majority voting, in order to build an optimal tree. The authors have also reduced communication costs between the node by introducing a parameter that controls how essential an event has to be in order for a node to send a message to its neighbouring nodes. Another parameter is introduced to enforce a minimum delay between the transmission of two consecutive messages on each node. An event could be the introduction or disappearance of a node in the network, the change of state of a node, or additional data received, that could possibly change the structure of the tree. Decision Trees and Random Forest are an interesting choice for real-time activity recognition as discussed in the previous section, thanks to their acceptable training and classification time, and generally good performance in the field. To the best of our knowledge, no fully distributed implementation of DT or RF have been used for activity recognition in a healthcare context.

Navia-Vazquez et al. [120] have addressed the distribution of the Support Vector Machine algorithm. They have presented a naïve approach where a local SVM is trained at every node using local data, the support vectors are sent to the other nodes, a new training set is built at each node using the support vectors received from other nodes, and new support vectors are computed and exchanged until convergence. Another method using Semiparametric Support Vector Machine is presented; it allows to reduce the communication costs compared to the naïve alternative. Both methods achieve a much better accuracy than a SVM only using locally available data on a single node. Support Vector Machine is not the fastest algorithm to train, but distributed optimizations such as the second method presented in that paper could allow dynamic retraining for a limited communication cost in sufficiently powerful WSNs.

In an embedded context, the main considerations are to keep processing as close as possible to the data source, to optimize sensor usage and scheduling in order to save energy, to compress data and promote efficient node communication, and to use a higher number of nodes if accuracy is the priority. The use of local optimization procedures is also mandatory to reduce communication costs, such as Fukushima’s local backpropagation for a distributed CNN [118].

### 5.4. Discussion

Real-time distributed activity recognition in a streaming context remains a very interesting field with many challenges to overcome. The six optimization angles of processing, memory, communication, energy, time, and accuracy leave no room for error in the conception of an efficient activity recognition system.

Different degrees of distribution can be implemented depending on the size of the WSN, as well as the algorithm and the context of the experiment. A higher number of nodes generally allows for a better accuracy through voting, and a better noise resilience, whereas a smaller number of nodes allows for less intrusive systems that can easily mesh in daily life activities. Processing can be reduced by using approximations, such as using integers instead of floats [106], extracting features with the highest information gain to extraction cost ratio [105], or downsizing the windows and the length of patterns to be matched [106], which also reduces the memory overhead necessary for gesture recognition. Optimizing sensors sampling rate allows both processing and memory savings [108]. Using integers also allows to save a lot of memory at the cost of a loss in accuracy. Compressing data using random projection is also a way to reduce memory usage as well as communication costs [110], but comes with extra on-node processing.

Communication can be optimized through the use of efficient protocols [109] as well as parameters controlling the importance of a message before sending it, as well as enforcing a minimum delay between consecutive transmission from the same node [119].

Energy is the variable that dictates most of the design and implementation choices for distributed activity recognition, especially in a healthcare context where it is expected to be able to monitor patient around the clock. Any processing optimization usually reduces energy consumption, whereas any extra processing to save on memory or communication will increase it. Rechargeable sensors can be a viable option in some cases [107]. In a sustainable development perspective, it would be interesting to include renewable, portable energy sources, such as small solar panels or energy harvesting modules based on body motion, whilst still making sure the system is as lightweight and non-intrusive as possible, especially in WBSNs.

Time is at the centre of real-time activity recognition, and any algorithm should be able to provide a classification result in a timely manner. Depending on the context of application, a one minute delay could be acceptable (activity of daily living monitoring) or it could be way too long (self-driving car).

The accuracy of the algorithm should be as high as possible despite all of these constraints, and a lot of time, a trade off has to be found between time and accuracy, as well as energy and processing. It is expected that distributed training based algorithm will not perform as good as good as their offline, batch training based equivalents, but it is a necessary compromise for real-time activity recognition. Instead, the trend goes toward higher degrees of distribution with dozens or hundreds of nodes to compensate the accuracy lost while training using partial data with an increase in data sources and a better modeling of the environment or the subject in dense WBSNs.

All of these angles are usually coupled, and trying to optimize one of them can result in a loss of performance in another one. A summary of some of the approaches used in the literature have been summarized in the table above (Table 4).

## 6. Conclusions

In this paper, we have reviewed activity recognition methods for healthcare in offline, centralized and distributed cases. We have highlighted different findings and challenges, and followed the natural evolution of activity recognition with the development of hardware performance, miniaturization, and the Internet of Things. When offline activity recognition acts as a sandbox for researcher to compare the performance of different machine learning algorithm and sensor types, real-time centralized methods allow real-life implementation for healthcare applications, but they come with a set of new problems such as efficient communication, online training and classification, real-time activity labeling, and concept drift detection. Real-time distributed activity recognition harnesses the power of the IoT to reduce communication, processing and memory cost, single points of failure and bottlenecks, dependence on an distant server, and an Internet connection. It allows us to promote pervasive computing and weaving of technology in everyday life to bring assistance and improve the subject’s quality of life.

Many improvements still have to be made to find balance between all the constraints that come with distribution of algorithms in a low resource environment. Research should focus on the six main optimization angles we have highlighted: Processing, memory, communication, energy, time, and accuracy. Various types of sensors and sensor combinations should be used for cases going from basic to complex activity recognition. Increasingly efficient and compact embedded algorithms have to be implemented for on-node processing. Smart and low energy communication protocols have to be used and improved, as communication is the crux of distribution. New energy sources have to be harvested. Existing machine learning algorithms have to be distributed in an efficient manner, especially deep learning and neural networks, which have shown a lot of potential. New distributed machine learning algorithms have to be crafted by switching from the monolithic nature of high volume batch learning to a more flexible and real-time perspective for training. New methods have to be explored to estimate the performance of a real-time system without ground truth. Semi-supervised as well as unsupervised methods have to be explored more thoroughly, as they will be more suited for real-time applications.

Beyond the scope of healthcare and activity recognition, distributed machine learning opens up a lot of opportunities both in research and industry for the implementation of distributed intelligence. By shifting the processing power away from the cloud and distant servers and bringing it closer to the data source, context is preserved, and quicker and more flexible systems can be developed. Each node is able to act here and now with local data in a case where communication is cut off, or collaborate with immediate neighbors, a subset of nodes in the network, or the entirety of its peers. It is easy to imagine an architecture with different tiers, degrees of distribution, accuracy, and complexity based on the task at hand for fields as different as surveillance, security, advertisement, industry, or even smart cities. Such a flexible system could also handle nodes joining and leaving the network and dynamically assign resources where they are needed. Real-time distributed artificial intelligence, embedded in the environment is a very challenging but promising field.

## Figures and Tables

**Figure 1 sensors-21-02786-f001:**
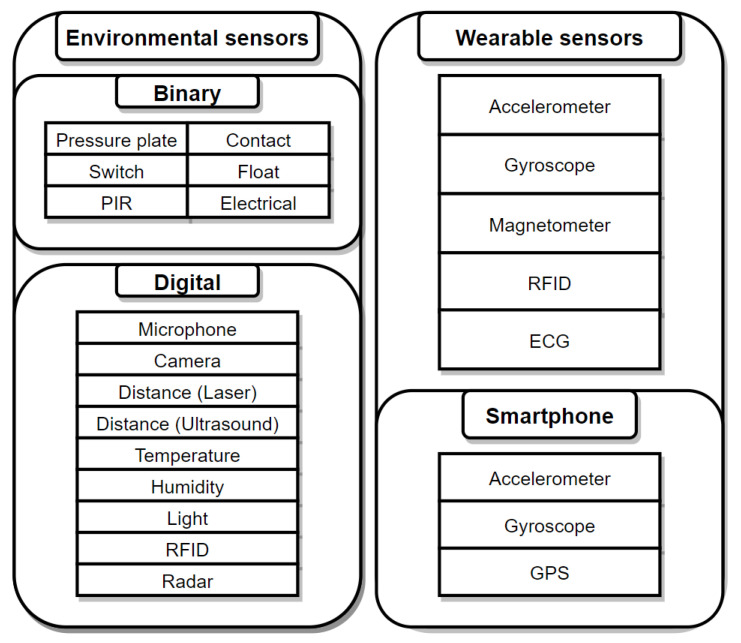
Overview of some of the reviewed sensors split into two main categories: Environmental and wearable sensors. Environmental sensors are divided into binary and digital sub-categories.

**Figure 2 sensors-21-02786-f002:**
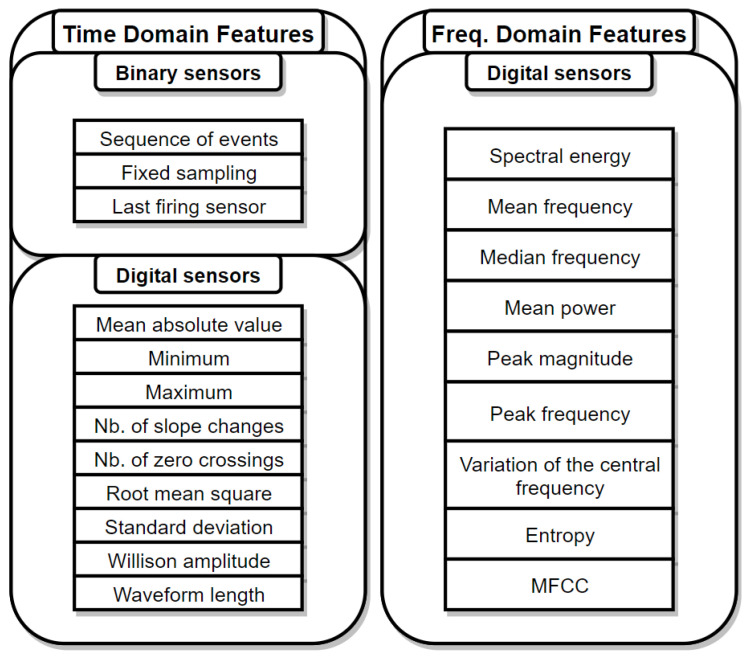
Overview of the most commonly extracted features split into time and frequency domain features. Features are further categorized based on the sensor used to collect the raw data.

**Figure 3 sensors-21-02786-f003:**
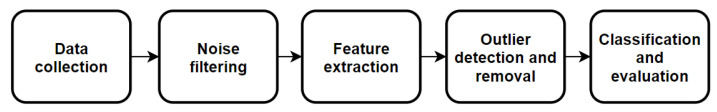
Pipeline of the standard activity recognition process, including the noise filtering and outlier removal phases.

**Figure 4 sensors-21-02786-f004:**
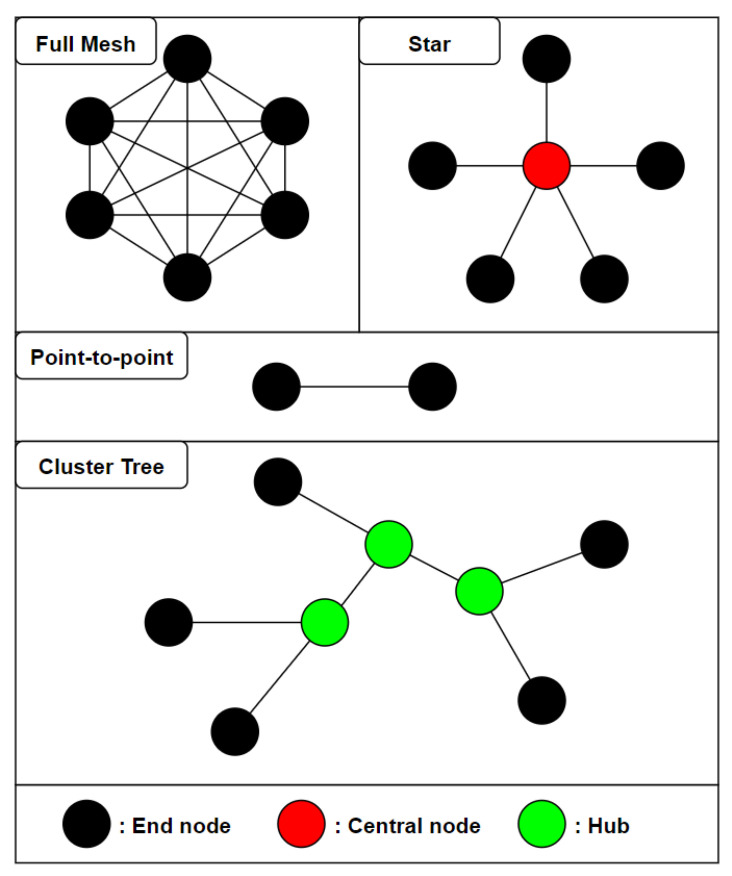
An overview of different network topologies used in wireless sensor networks for real-time activity recognition. In the full mesh (**top left**), any node can directly communicate with any other node in the network. In the star topology, all of the end nodes are connected to a single central node. In a point-to-point topology (**middle**), two nodes communicate strictly with one another. In a cluster tree (**bottom**), end nodes are connected to different hubs which are connected to each other.

**Figure 5 sensors-21-02786-f005:**
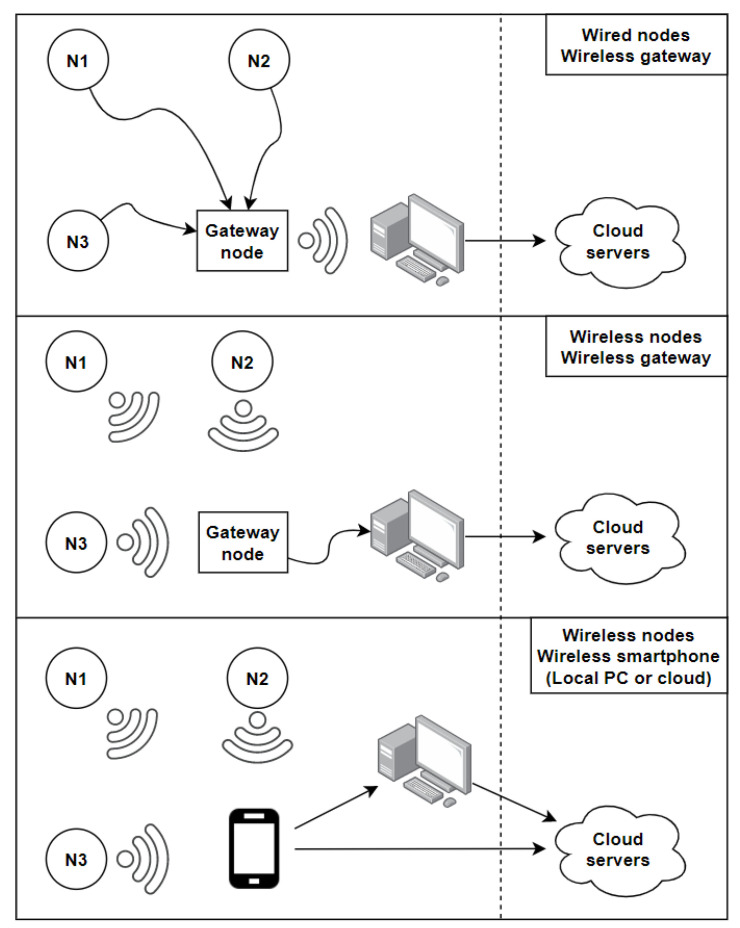
Diagram representing the three main local and cloud architectures used for real-time centralized activity recognition. Sensor nodes can be connected to a gateway node using USB/RS232 cables. The gateway node sends the collected data wirelessly to a central station that can process the data locally or send it over to a cloud (**top**). In the second configuration, sensor nodes can send the collected data wirelessly to a gateway node, connected to the central station (**middle**). In the third configuration, all communication is performed wirelessly between the sensor nodes, a smartphone used as a gateway node, and a central station or cloud servers (**bottom**).

**Figure 6 sensors-21-02786-f006:**
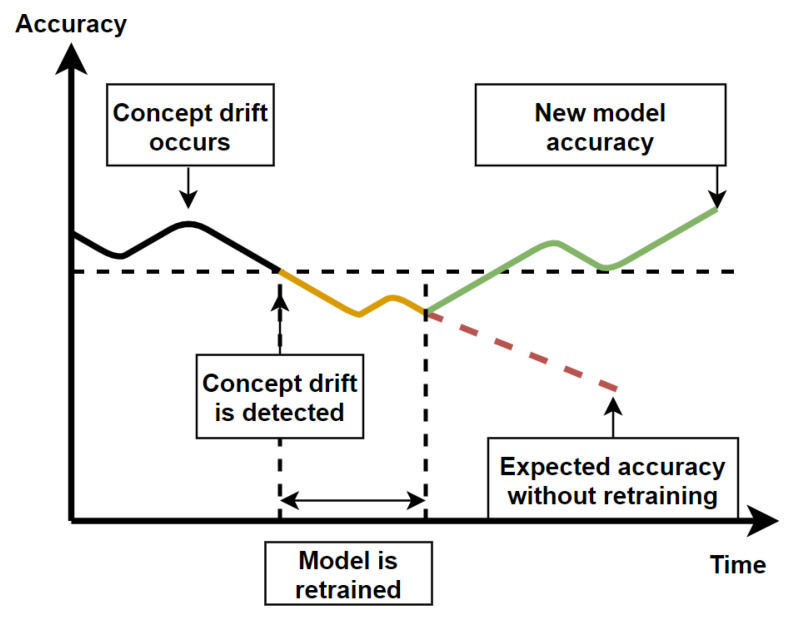
Diagram of the evolution of a model’s accuracy over time as concept drift occurs in two cases: With retraining and without retraining.

**Figure 7 sensors-21-02786-f007:**
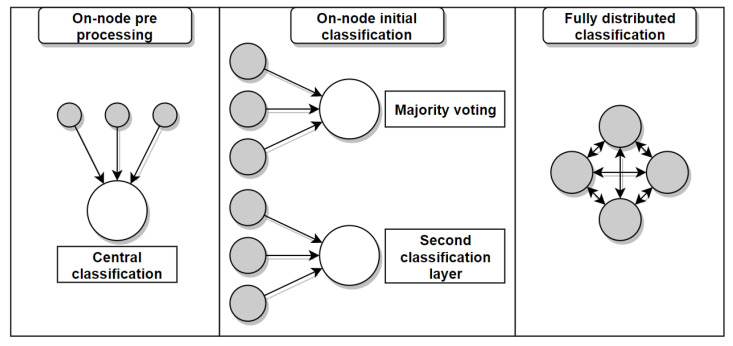
Diagram of different degrees of distribution for distributed machine learning in WSNs. On-node pre-processing is performed before sending the data to a central node for classification (**left**). Initial classification is performed on the sensor nodes before performing a second classification on the central node (**middle**). The classification process is distributed among the nodes (**right**).

**Table 1 sensors-21-02786-t001:** Comparison table for offline environmental sensor-based activity recognition approaches.

Paper	Sensors	Features	Window Size	Algorithms	Activity Recognition Accuracy
[47]	Object accelerometers	TDF	2 s	DT, MLP, SVM	78.9% (SVM), 78.4% (MLP), 74.7% (DT C4.5)
[52]	Microphone	MFCC	1.5 s	DTW	92.5%
[25]	AICO	TDF	Var.	Bay. Net.	80%
[49]	PIR, switch, float, pressure, contact, temperature	Raw data, change point, last firing sensor	1 min	HMM, CRF	95.1% (CRF), 91.2% (HMM)
[19]	PIR	Seq. of activation	1 min	HMM/MLP, HMM/SVM, HMM	67.2% (HMM/SVM), 65.4% (HMM/MLP), 52.2% (HMM)
[22]	PIR, reed switches, pressure, contact, float	Sensor data vs contextual info.	1 min	FCA	93.8% (Sensor data), 85.6% (Context)
[21]	Motion, pressure	Change of state	10 min	HMM	85%
[30]	COTS Radar	Bandwidth, min, max	N/A	PCA	N/A
[24]	FSR, photocells, distance, sonar, contact, temp, PIR, pressure	Seq. of activation	1 min	k-NN, DT, MLP, TDNN, HMM	71.8% (TDNN), 65.2% (DT), 64.4% (MLP), 63.6% (kNN), 62.2% (HMM)
[50]	CASAS dataset	Per-day act. distribution	5 min	DT	80%
[23]	PIRs with mask	Short term energy	2 s (50% OL)	2-layer RF	82.5% (First layer), 92.5% (Second layer)
[20]	Binary sensors	Influence of each sensor	N/A	NB, HMM, CRF	79% (CRF), 76.6% (HMM), 71% (NB)
[28]	Microphone	TDF	10 s	RF	95%
[27]	RFID tags on objects	Object state	Var.	LSTM	85.7%

**Table 2 sensors-21-02786-t002:** Comparison table for offline wearable sensor-based activity recognition approaches.

Paper	Sensors	Features	Sampling Frequency	Window Size and Overlap	Algorithms	AR Accuracy
[41]	Accel. Env Sensors	TDF, FDF (FFT)	N/A	50% OL	GMM + FSM	93.9%
[34]	3 accel. 2 RFID WR humidity sens.	TDF, temp humidity, location	N/A	15 s	epSICAR	91% (seq.) 88% (inter.) 78.6% (conc.)
[45]	ECG and accel.	TDF	300 Hz (ECG) 75 Hz (accel)	0.12 s (ECG) 0.48 s (accel) 50% OL	SVM, GMM	84.8% (SVM) 79.7% (GMM) 89.3% (Mix)
[40]	2 accel. 2 RFID WR mic, humidity, temp, light	TDF FDF for mic Location (RFID)	128 Hz (accel) 2 Hz (RFID)	1 s	CHMM, FCFR	96.4% (CHMM) 87.9% (FCRF)
[37]	2 accel.	TDF	10 Hz	50% OL	BT + NN	99.2%
[36]	3 accel. and gyr.	TDF	150 Hz	N/A	KM + HMM	90.2%
[33]	3 accel.	TDF and FDF (ST and FFT)	90 Hz	1 s 87% OL	SVM, DT	96% (SVM) 90% (DT)
[42]	Phone accel.	TDF	50Hz	2 s	ANN	93%
[43]	Phone accel.	TDF and FDF	50 Hz	2.56 s 50% OL	SVM	96.6%
[44]	Accel. gyr. and magnetometer	TDF	5 Hz 25 Hz (phone)	2 s	k-NN, ANN, SVM, CART	96.8% (ANN) 96.2% (k-NN) 95.3% (CART) 94.4% (SVM)
[58]	5 IMU 12 accel	N/A	30 Hz	0.5s 50% OL	DeepConv- LSTM	86.6%
[39]	4 accel.	TDF	N/A	1 s 15% OL	SRC-RP, SVM, HMM	94% (SRC-RP) 87% (SVM) 82% (HMM)
[31]	Accel. gyr. and magnetometer	TDF and FDF	125 Hz	0.5/1/2 s 0.25s OL	MLP	91.7% (TDF) 88.5% (FDF)
[59]	Accel.	TDF and FDF	50–100 Hz	50% OL	SVM, k-NN	95.5% (SVM) 96.7% (k-NN)
[46]	ECG and accel.	TDF and FDF	500 Hz (ECG) 50 Hz (accel)	1.28 s	DT	96.92%
[55]	Accel.	TDF and FDF	20 Hz	10 s	SVM, RBFN	91.4% (SVM) 86.2% (RBFN)
[35]	RFID bracelet	TDF	N/A	20 s	Rule-based classification	88%

**Table 3 sensors-21-02786-t003:** Comparison table for Wi-Fi, Bluetooth, BLE, ANT, and ZigBee standards in terms of speed, range, energy consumption, compatible topologies, and maximum number of nodes in a single network. The highest values in have been highlighted in the speed, range, and max number of nodes category, and the lowest value in the energy consumption category.

Protocol	Speed (Mbps)	Range (m)	Energy cons. (mW)	Topologies	Max Nodes
Wi-Fi	**1300**	90	12.21	P2P, Star	250
Bluetooth	24	100	4.25	P2P, Broadcast	7 (active)
BLE (5.0)	2	240	**0.07**	P2P, Broadcast, Mesh	7 (active)
ZigBee	0.25	100	0.66	P2P, Star, Cluster Tree, Mesh	**65,536**
ANT	0.06	30	0.83	P2P, Star, Tree, Mesh	65,533
LoRaWAN	0.027	**19,000**	1.65	Star of stars	120

**Table 4 sensors-21-02786-t004:** Comparison of the impact of different optimization methods used in the literature on processing, memory, communication, energy, time, and accuracy of the system. A *+* symbol (green cell) means a positive impact, a *=* symbol (yellow cell) means no noticeable impact, a *-* symbol (red cell) means a negative impact, and a ∼ symbol (gray cell) means an impact that could be positive or negative based on the method implementation.

Method	Processing	Memory	Comm.	Energy	Time	Acc.
Integers instead of floats	+	+	+	+	+	-
Highest gain features	+	=	=	=	=	-
Smaller windows/patterns	+	+	+	+	+	-
Lower sensor sampling rate	+	+	+	+	=	-
Compressing data on node	-	+	+	∼	=	-
Comm. reduction protocols	-	-	+	∼	-	-
Rechargeable sensors	-	=	-	+	=	=
Sensor subselection	-	=	+	+	-	-
More nodes	-	-	+	-	=	+
Majority voting	=	=	+	+	=	+
2-layer classification	+	=	-	-	=	+
Local optimization	+	+	+	+	+	-

## Data Availability

Not applicable.

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
