# Peer review of "From Offline to Real-Time Distributed Activity Recognition in Wireless Sensor Networks for Healthcare: A Review"

_sensors, 2021, doi:10.3390/s21082786_

Round 1

Reviewer 1 Report

This is a well-written and detailed paper that surveys human activity recognition for medical purposes in offline, centralized, and distributed scenarios. The authors did a good job with presenting a comprehensive study and categorizing things and a nicely-flowing and useful manner. I am fine accepting this paper in its current condition.

Author Response

See attached pdf file

Reviewer 2 Report

I think this review is a confusing collection of superficial techniques. A lot of techniques are covered in the manuscript, including sensors, labeling, feature extraction, filtering, recognition algorithms, communication techniques, network topology, machine learning, etc. These technologies are generalized, without in-depth analysis of their strengths and weaknesses.

This article failed to introduce the related published reviews&surveys. Therefore, it is difficult to evaluate the value of this paper. It is recommended to elaborate on the contributions.

Section II focuses on offline activity recognition, but the title is “real-time activity recognition”, which is obviously unfair.

It is recommended that only two to three key techniques are discussed in depth. The purpose of the review article is to provide readers with detailed research status and comparative analysis of advantages and disadvantages of related technologies, and to summarize the current challenges faced by each key technique. Potential solutions to each challenge are also need to be presented.

Author Response

See attached pdf file

Reviewer 3 Report

The authors should use PRISMA. Moreover, the Research Questions needs to be presented in a systematic way.  The analysis does not present relevant findings.

A lot of important databases are not included such as IEEE Xplore, PUBMED, Web of Science and ACM Digital Library. Why the authors only use Scopus? 

The IC (inclusion criteria) and EC( Exclusion Criteria) should be clearly presented and justified. "Any paper was considered relevant if it mentioned human activity recognition using sensors." - this is not enough to justify the selection.

What is the specific query string to get the results and reproduce the search process?

Who and how have selected the studies?

Why this research is necessary? How this study will support future research activities in this field?

The authors need to compare this review with other reviews in this field. Moreover, the limitations of the review study and the process should be stated.

I recommend the authors to check "http://prisma-statement.org/".

Author Response

See attached pdf file

Reviewer 4 Report

The review is very interesting and extremely topical. I believe that the quality of the work is adequate. However, I suggest some improvements that I believe need to be implemented.

*) Please make the captions in the Figures self-explanatory.

*) There are some typos. Please delete them.

*) Figure 5 is very interesting. Please comment on it more by offering food for thought for future developments.

*) There are some (??) instead of bibliographic references. Please recompile the document.

*) Many acronomics are present in the text. Perhaps it would be appropriate to insert a table that lists them.

Author Response

See attached pdf file

Round 2

Reviewer 2 Report

The authors reviewed activity recognition methods for healthcare in offline, centralized and distributed cases. It is including machine learning, distributed, embedded real-time and wireless data transmission on different queries. The paper focused on human health applications, which is meaningful and practical significance.

As far as we known, the smart IOT devices are widely used in industrial, agricultural and medical fields. Especially on human-care, so many AI methods were presented in three years, like sportsman monitoring, baby care and smart Elderly system. Therefore, I have some suggestions:

  1. Please update the references. At least published in five years.
  2. Please focus on your research work. Or introduce why you want to review it. Otherwise, I can’t find the contribution.
  3.  In WSN, the Wi-Fi, Zigbee and LoRaWAN technologies have limitations by protocols.  Whether the new communication technologies are applicable for healthcare? For example, 5G/6G network, IEEE 802.11af protocol ,700MHz broadband communications and so on. 

Author Response

See attached pdf file

Reviewer 3 Report

Thank you for your revision. The document is improved. 

However, the structure is an issue right now.

A lot of descriptions about research questions, methods for screening, inclusion and exclusion criteria should be removed from the introduction and new section should be added denominated "Methods".

Author Response

See attached pdf file
